# Post-Operative Modified All-Inside ACL Reconstruction Technique’s Clinical Outcomes and Isokinetic Strength Assessments

**DOI:** 10.3390/diagnostics13172787

**Published:** 2023-08-29

**Authors:** Ahmet Serhat Genç, Nizamettin Güzel, Ali Kerim Yılmaz, Egemen Ermiş, Mine Pekesen Kurtça, Anıl Agar, Kubilay Uğurcan Ceritoğlu, Yavuz Yasul, İsmail Eseoğlu, Lokman Kehribar

**Affiliations:** 1Department of Orthopedics and Traumatology, Samsun Training and Research Hospital, Samsun 55100, Türkiye; ahmetserhatgenc@hotmail.com (A.S.G.); nizamettin.guzel@saglik.gov.tr (N.G.); 2Faculty of Yasar Dogu Sport Sciences, Ondokuz Mayıs University, Samsun 55100, Türkiye; egemen.ermis@omu.edu.tr; 3Faculty of Health Science, Ondokuz Mayıs University, Samsun 55100, Türkiye; mine.pekesenkurtca@omu.edu.tr; 4Department of Orthopaedics and Traumatology, Fırat University, Elazığ 23119, Türkiye; dr.anilagar@hotmail.com; 5Department of Orthopaedics and Traumatology, Onsekiz Mart University, Çanakkale 17100, Türkiye; kubilay.ceritoglu@comu.edu.tr; 6Bafra Vocational School, Ondokuz Mayıs University, Samsun 55400, Türkiye; yavuz.yasul@omu.edu.tr; 7Vocational School of Health Services, Dokuz Eylül University, İzmir 35210, Türkiye; ismaileseoglu@hotmail.com; 8Department of Orthopaedics and Traumatology, Samsun University, Samsun 55090, Türkiye; lokmankehribar@gmail.com

**Keywords:** anterior cruciate ligament reconstruction, return to sports, isokinetic evaluations, modified all-inside, knee strength, knee scores, surgical side

## Abstract

Background and Objective: Anterior cruciate ligament (ACL) injuries are very common among the athletic population. ACL reconstruction (ACLR) performed because of these injuries is one of the procedures performed by orthopedic surgeons using different grafting methods. This study aims to compare the data related to post-operative 6-month isokinetic strength values, strength-related asymmetry rates, time parameters, and joint angle in athletes who underwent ACLR with the Modified All-inside (4ST) technique, on both the healthy knee (HK) and the ACLR-applied sides. Materials and Methods: A total of 20 athletes from various sports on whom the 4ST ACLR technique had been applied by the same surgeon were evaluated retrospectively. Lysholm, Tegner, and International Knee Documentation Committee (IKDC) scores of the patients were obtained pre-operative and at 6 months post-operative. Isokinetic knee extension (Ex) and flexion (Flx) strengths on the HK and ACLR sides of the patients were evaluated with a series of four different angular velocities (60, 180, 240, and 300°/s). In addition to peak torque (PT) and hamstring/quadriceps ratio (H/Q) parameters, the findings were also evaluated with additional parameters such as joint angle at peak torque (JAPT), time to peak torque (TPT), reciprocal delay (RD), and endurance ratio (ER). Results: There was a significant improvement in the mean Lysholm, Tegner, and IKDC scores after surgery compared with pre-operative levels (*p* < 0.05). As for PT values, there were significant differences in favor of the HK in the 60, 180, and 300°/s Ex phases (*p* < 0.05). In terms of the H/Q and (hamstring/hamstring)/(quadriceps/quadriceps) (HH/QQ) ratios, there were significant differences at 300°/s (*p* < 0.05). In terms of JAPT, there were significant differences in the 300°/s Ex and 180°/s Flx phases (*p* < 0.05). In terms of TPT, there were significant differences in the 300°/s Ex phase (*p* < 0.05). In terms of RD and ER, no significant difference was observed between the HK and ACLR sides at any angular velocity. Conclusions: Although differences were observed in PT values, particularly in the Ex phase, this did not cause a significant change in H/Q ratios. Similar results were observed for additional parameters such as JAPT, TPT, RD, and ER. The results show that this ACLR technique can be used in athletes in view of strength gain and a return to sports.

## 1. Introduction

The anterior cruciate ligament (ACL) plays an important role in the stabilization of the knee joint and the motor control mechanism of the knee [1]. ACL reconstruction (ACLR) is one of the most common orthopedic surgical procedures performed by sports physicians since ACL is one of the most common injuries suffered by athletes [2,3,4]. It has been reported that the risk of ACL injury is higher in competitive women and young athletes aged 15–25 years [5,6]. ACL is particularly important for people with high physical activity because it adjusts the stiffness of the quadriceps (Q) and hamstring (H) muscles, which form the agonist–antagonist structure of the knee [7]. Following ACL injuries, mechanical, proprioceptive, and efferent neuromuscular disorders, decreased muscle function and strength, and an imbalance in strength–torque generation may be observed [8,9]. ACL injuries may occur as a result of sudden deceleration and direction changes, sprinting and hitting the ball, or hard blows to the knee [1,10,11]. Different methods such as quadriceps tendon, patellar tendon, and semitendinosus/gracilis (ST/G) tendon are applied by physicians in ACLR [12]. In addition to these techniques, the “all-inside” technique, which uses only four-fold (4ST) grafting with ST has been frequently used recently [13]. Since the all-inside technique has several disadvantages despite its advantages, a modified all-inside (MAI) technique was developed by Mahirogullari et al. [14]. In the MAI technique, the ST graft, which is prepared by folding the ST tendon four times, is fixed to the tibia and femur with suspension [15]. Although the MAI technique shares many of the advantages of the all-inside technique, such as using a single tendon and creating a socket instead of a tunnel on the tibial side, it is free from the unmodified technique’s disadvantages of the need to use special burs on the tibial side, the creation of the socket, the limited margin of error in the adjustment of the socket depth, the need to place the graft from the portal, and the higher cost [16].

ACLR aims to restore the knee joint function and stability while the rehabilitation process seeks to restore the neuromuscular losses in the Q and H muscles [17,18]. The rehabilitation process is extremely important in making the decision about a return to sports (RTS), particularly for the athlete population [8]. Although many methods are used for determining if an athlete is ready for a RTS, the most objective method with regard to measuring strength is to use isokinetic dynamometers [15,19,20]. Traditionally, isokinetic dynamometers use flexion/extension (Flx/Ex) or H/Q ratios, which are formed by the strength produced by Q and H muscles in the extension (Ex) and flexion (Flx) phases of the knee [20,21]. After ACLR, imbalances in H/Q ratios can be observed due to the asymmetrical strength in the lower extremity. The H/Q ratio may vary between 50% and 80% as the angular velocity in the dynamometer increases [22,23]. For an angular velocity of 60°, a H/Q ratio of 2/3 can be considered normal [22,24].

Thanks to the isokinetic dynamometers, not only the conventional data (PT and H/Q ratio) but also the connection angle Joint Angle at Peak Torque (JAPT), which shows the peak torque reached, the time loss in the reciprocal transitions between the phases Reciprocal Delay (RD), the times to reach the peak torque Time to Peak Torque (TPT) in the Ex and Flx phases, and data such as endurance indices Endurance Ratio (ER) occurring in the Q and H muscles occurring at 15 or more repetitions can also be collected [3]. Researchers have emphasized the importance of muscle reaction times during strength generation, particularly in preventing musculoskeletal injuries in joints [25,26]. Zabka et al. [27] pointed out that delays in reaction times of agonist–antagonist muscles might cause serious knee injuries, particularly in athletes who frequently make sudden speed and direction changes.

Although peak torque (PT) and H/Q values are emphasized in numerous studies conducted for the evaluation of post-ACLR processes, the number of studies evaluating additional objective data provided by isokinetic tests such as JAPT, TPT, RD, and ER is limited. The aim of the present study was to evaluate knees that underwent ACLR with the MAI technique and healthy knees using additional parameters obtained with isokinetic tests in addition to traditional data such as PT and H/Q. The hypothesis is that the MAI technique will produce similar results in the ACLR-applied side to the healthy side in post-operative knee scores, parameters that occur due to traditional and additional strength.

## 2. Materials and Methods

### 2.1. Patients

The present study was conducted using retrospective records of patients treated for ACL rupture in the orthopedics and traumatology department of an academic medical center. Ethics committee approval was provided by Samsun University, Clinical Research Ethics Committee (Approval Number: SÜKAEK-2023-2/8), and the study was conducted in accordance with the Declaration of Helsinki. All participants were informed about the study and informed consent forms were obtained. The study included a retrospective cohort of recreative athletes (*n* = 20) who underwent MAI ACLR by the same surgeon between January 2020 and December 2021. An a priori test with the GPower 3.1 program was used to determine the number of participants. As a result of the sample study we took for power analysis, it was calculated that the study could be completed with 18 patients (Effect size: 0.82, Actual Power: 0.93) [15]. The data were collected in the Ondokuz Mayıs University Yaşar Doğu Faculty of Sport Sciences Performance Laboratory in Samsun, Türkiye. All measurements took approximately 2 years to complete.

The inclusion criteria for the study were as follows: being a male between the ages of 18 and 35 with isolated ACL rupture in only one knee and without any concomitant meniscus, chondral, or other ligament injury, and having no other neuromuscular or musculoskeletal injury or a history of contralateral knee surgery or injury. Lysholm, Tegner, and International Knee Documentation Committee (IKDC) scores of the patients were evaluated before and at the 6th month post-operatively. The Lysholm knee score is a condition-specific outcome measure that includes eight domains. An overall score of 0 to 100 points is calculated in this test, and a score of 95 to 100 indicates an excellent result [28,29]. The Tegner Activity Scale was designed as an activity level score to complement the functional Lysholm knee score in patients with ligament injuries [30]. The scale scores a person’s activity level from 0 to 10; where 0 means “on sick leave/disability” and 10 means “participation in national or international elite competitive sports such as football” [31]. The IKDC subjective score consists of 18 items addressing symptoms, sports activities, and functional performance. It comprehensively covers an individual’s health status and encompasses simple questions such as support, locking, and stair climbing [32]. In order to reduce variability in recovery periods, all participants were referred to the same rehabilitation program after surgery. The average time between the injury and the surgery was about 2 months. Detailed characteristics of the cohort are shown in Table 1.

### 2.2. Surgical Treatment

Anatomical single-bundle ACLR was applied to all patients using adjustable suspension fixation with a quadruple semitendinosus tendon autograft (Figure 1). The removed semitendinosus tendon (24–28 cm long) was quadrupled with adjustable loop cortical suspend fixation (Lift Loop External; Orthomed, Ankara, Türkiye) on the tibial end and with a fixed loop button system (Femobutton; Orthomed, Ankara, Türkiye) on the femoral side. The Lift Loop External fixation system consists of a 20 mm-wide titanium button and 2 loops controlled by a knotless locking mechanism. Femobutton consists of a 10 mm-wide titanium button and a continuous loop available in 6 lengths (15–40 mm). The anatomical femoral tunnel was carved from the anteromedial portal. Firstly, a full bone tunnel was drilled over the guide pin using a 4.5 mm drill. Then, the tunnel length was measured and a socket was opened using a router the same size as the graft, taking into account the 6 to 8 mm endobutton ‘flip’ movement distance. Afterward, a complete outside-in tibial tunnel was created at the central location of the anatomical footprint. The graft was inserted from the intra-articular space to the tibial and femoral tunnels. The graft was stretched at 20° flexion of the knee. The knee was flexed and extended 30 times. Graft tightness was examined with a probe. Finally, the entire structure was re-tensioned on the tibial side and additional tethering was done on the adjustable suspension fixation device using a non-slip knot [14]. The materials used in post-operative roentgenography are presented in Figure 2.

### 2.3. Experimental Approach of the Study

Lysholm, Tegner, IKDC scores (pre- and post-operative) and 6-month post-operative isokinetic knee Ex and Flx performances of all participants were determined. For all tests conducted, the patients visited the laboratory 3 times in total, except for the routine controls before and after ACLR. In the first visit (pre-operative), the participants were asked to fill in the subjective questionnaires, which included Lysholm, Tegner, and IKDC scales, and were informed about the study. In the second visit (6 months post-operative), anthropometric data were collected and isokinetic knee strength tests to be performed in the following visit were experienced by the participants (familiarization). In the third laboratory visit (2 days after the second visit), Lysholm, Tegner, and IKDC scales were filled for the second time (post-operative), and 6-month post-operative isokinetic knee Ex and Flx performances were measured.

### 2.4. Procedures (Isokinetic Strength Measurement)

The knee Ex and flexion Flx strengths of the participants in the healthy knee (HK) and ACLR-applied knee sides were evaluated with a series consisting of 4 different angular velocities (60, 180, 240, and 300°/s) (Figure 3). A computer-controlled isokinetic dynamometer (Humac Norm Testing and Rehabilitation System, CSMI, Stoughton, MA, USA) was used for this evaluation. Once the general warm-up protocol [33] was completed, the seat, dynamometer, adapter, and other settings of the dynamometer were adjusted for the subjects according to the fixed protocol set for knee Ex and Flx strengths [22]. In accordance with this protocol, the mobility angle (range of motion (ROM)) of the subjects’ knee joints was fixed to a 0–90° position. The back support of the chair was set at the hip joint angle of 85° (0° = full extension). Dynamometer arm rotation was set at the level of the lateral femoral epicondyle. The pad on which the lower leg attachment was fixed was positioned proximal to the lateral malleolus. The belts that prevented body and Q muscle movement were tightened leaving a three-finger gap between the body and the Q muscle, and subjects held the hand grips on both sides of the seat during the test. The ankle was positioned on the leg stabilizer under the chair to prevent movement of the contralateral limb. Prior to the tests, the knee joint rotation axis (lateral femoral condyle) and rotation axes were calibrated on the same line. Before the measurements, the torque value of the knee joint produced by the leg at 90° Ex (full extension) in the free position was measured with a dynamometer in all subjects to eliminate the effect of gravity. This ensured that the torque values collected with the measurements were only strength-based torque values. Before starting the test, all subjects were asked to apply their knee strength at a maximum level to achieve a positive test and to obtain maximum results. 

Knee Ex and Flx strengths for both HK and ACLR sides were measured by adjusting the fixed protocol performed with sequential concentric/concentric (Con/Con) contractions at angular velocities of 60°/s (4 retries, 15 s rest, 5 retests), 180°/s (4 retries, 15 s rest, 5 retests), 240°/s (4 retries, 15 s rest, 15 retests), and 300°/s (4 retries, 15 s rest, 15 retests). One-minute rest intervals were given between angular velocities, and five-minute rest intervals were given between ACLR and HK sides. The tests were first applied to the ACLR side. In order to achieve maximal results, verbal support was given to the subjects throughout the measurements to increase motivation. Peak torque (PT) values were recorded in Newton meters (Nm); H/Q, H/H, and Q/Q ratios were recorded in percentage (%); JAPT values were recorded in degrees (°); TPT and RD values were recorded in seconds (s); and ER values were recorded in Nm. Since ER values are obtained only in tests performed with 15 repeats or more in isokinetic dynamometers, these values were collected only at angular velocities of 240°/s and 300°/s [8].

### 2.5. Statistical Analyses

SPSS 21 (IBM Inc., Chicago, IL, USA) was used in the statistical analysis of the study. Results were presented as mean, standard deviation, minimum, maximum, and median. The Shapiro–Wilk test was used for normality testing and Levene’s test was used for homogeneity assumptions. A paired sample test or the Wilcoxon test was used to compare paired groups (HK–ACLR and pre–post). The “*t*” values obtained in the research include the paired sample *t*-test results applied between the ACLR–HK side and pre–post measurements. For the comparison of three or more groups, the ANOVA test was used for the internal analysis of H/H and Q/Q ratios at all angular velocities. All 95% CI values in statistical comparisons include pre–post test results. In addition, in the comparison of paired groups, effect sizes were found according to Cohen’s *d* effect size (M2 − M1)⁄SDpooled). According to this formula, a *d* value of <0.2 was defined as a weak effect size, a *d* value of 0.5 was defined as a moderate effect size, and a *d* value of >0.8 was defined as a strong effect size. The statistical results were evaluated within a significance level of *p* < 0.05.

## 3. Result

Table 1 shows the mean age (28.05 y), height (176.75 cm), weight (86.40 kg), BMI (27.58 kg/m^2^), and Time from ACLR to measurements (6.95 mo) of the patients. The dominant sides of the patients were recorded as 15 right, 5 left, and the surgical side of the patients as 14 right and 6 left.

There were significant differences between pre- and post-operative Lysholm (*p* < 0.001), IKDC (*p* < 0.001), and Tegner (*p* < 0.001) scores, as presented in Table 2.

Table 3 shows a comparison of the knee peak torque and H/Q ratios revealed by isokinetic tests at different angular velocities on the ACLR and HK sides. The findings showed that peak torque was significant at 60°/s Ex (*p* = 0.008), 180°/s Ex (*p* = 0.024), and 300°/s Ex (*p* = 0.005). As for the H/Q ratios, there was a significance only at 300°/s (*p* = 0.002).

Table 4 presents the H/H and Q/Q ratios obtained at different angular velocities in the ACLR and HK sides in isokinetic tests. Accordingly, there was a statistically significant difference in the 300°/s (*p* = 0.002, ES = 0.88) parameter between the HK and ACLR sides. Although there was no statistically significant difference in the 60°/s (ES = 0.88), 180°/s (ES = 0.88), and 240°/s (ES = 0.88) parameters, intermediate-level effect sizes were determined (Table 4).

There was significance at 300°/s Ex (*p* = 0.013) and 180°/s Flx (*p* = 0.048) for JAPT values obtained at different angular velocities in isokinetic tests (Table 5).

Table 6 shows the comparison of TPT values obtained in isokinetic tests for ACLR and HK sides. The results revealed that there was significance only at 300°/s Ex (*p* = 0.015).

The comparison of RD values obtained for ACLR and HK sides at different angular velocities in isokinetic tests revealed statistically similar results (*p* > 0.05) (Table 7).

Table 8 shows the comparison of ER values obtained for ACLR and HK sides at different angular velocities in isokinetic tests. ACLR and HK sides produced similar results (*p* > 0.05).

## 4. Discussion

In this study, the relationships between pre- and post-operative knee scores (Lysholm, Tegner, IKDC) of ACLR-applied knees and healthy knees in athletes who underwent ACLR with the MAI technique were evaluated with regard to several data obtained with isokinetic tests. The comparisons conducted in the study revealed that there were significant differences between the ACLR-applied knee and the healthy knee at 60°/s, 180°/s, and 300°/s Ex for PT values; at 300°/s for H/Q and H/H and Q/Q ratios; at 300°/s Ex and 180°/s Flx for JAPT values; and at only 300°/s Ex for TPT values. Although the results of the study revealed differences between the ACLR-applied knee and the healthy knee regarding certain data, in general, similar results were obtained in many parameters between the operated-on knee and the healthy knee. 

Objective data related to knee strength are commonly obtained through isokinetic dynamometers in order to make a decision with regard to a RTS and monitor the rehabilitation process after ACLR [19,20,21]. Post-ACLR isokinetic dynamometers provide objective data on PT values, particularly in Ex and Flx movements of the knee and Q and H muscles [34]. With these data, H/Q ratios at different angular velocities and asymmetries between the limbs are evaluated. In their study which evaluated the pre- and post-operative knee strengths of athletes who underwent ST/G ACLR, Riesterer et al. [35] found similar strength increases between the ACLR and HK groups in the Flx phase although the strength increase was higher in the ACLR group in the Ex phase. Güzel et al. [8] conducted a study evaluating the pre- and post-operative knee strengths of athletes who underwent ST/G ACLR and similarly found that there was a higher increase in strength in the Q muscle group compared to H and that this did not cause a significant difference in H/Q ratios except for at 60°/s. In the literature, there are both studies comparing ACLR-applied knee and healthy knee before and after the operation and studies comparing different ACLR techniques. In one of these studies, Roger et al. [36] examined pre- and post-operative knee strengths between ST/G and 4ST ACLR groups and found that the 4ST group showed better recovery compared to the ST/G group, although there was no statistically significant difference between the two techniques. Cavaignac et al. [37] conducted a study comparing the knee strengths of ST/G and quadricep tendon (QT) ACLR-applied groups and found similar results in Ex, Flx, and H/Q ratios for the groups. Although significant differences were found between the ACLR-applied knee and the HK regarding PT values, particularly in the Ex phase, in the present study, no significant difference was observed between the H/Q ratios in these two groups, which indicates that the findings of our study are similar to those of the literature. 

Although traditional methods such as PT and H/Q are frequently used in studies involving post-ACLR evaluations in the literature, the number of studies based on objective data such as JAPT, TPT, RD, and ER obtained with isokinetic dynamometers is limited. In their study involving JAPT, TPT, and RD values in addition to PT and H/Q values, Genç & Güzel [3] compared operated-on knees and HKs in the post-ST/G ACLR period and found similar results with regard to these values. Although the ACLR technique applied in the present study was different, JAPT, TPT, RD, and ER values were examined in addition to PT and H/Q values, and similar results were obtained in general. Researchers consider the JAPT value as an important indicator of the risk for muscle injuries as it indicates the relationships between muscle length and tension [38,39]. The importance of data related to muscle reaction time such as TPT and RD in determining joint injury risks after sudden movements is also emphasized [40,41].

MAI is considered an advantageous ACLR technique since it provides many advantages such as the use of a tendon and the creation of a socket instead of a tunnel on the tibial side as in the 4ST technique. It also lacks many of the disadvantages inherent in the 4ST technique such as the necessity of using a special burs, the difficulties in creating the socket and adjusting its depth, the necessity of placing the grafts through the portal, and a high cost [14,16]. The present study is the first study evaluating isokinetic test parameters after MAI ACLR. In addition, it is one of a limited number of studies in the literature that examines JAPT, TPT, RD, and ER values in addition to PT and H/Q values after ACLR. It is important to consider isokinetic data such as JAPT, TPT, RD, and ER in making the RTS decision and deciding rehabilitation processes after ACLR in order to better understand these processes. 

Our research had some limitations in order to evaluate the results clearly. Our main limitations are that the study was conducted only on male patients and healthy controls were not used. In addition, although our patients were athletes, our other limitations were that they were interested in different sports and had different training ages. Finally, in our study, there were only patients who underwent the MAI ACLR technique. Thus, future studies should include repeated measurements, comparison of different ACLR techniques, and evaluation of the effects of ACLR on different activity levels and genders. This will enable more precise results about the post-ACLR processes. Similarly, the comparison of the MAI technique, which has not been sufficiently studied yet, with various techniques using different parameters will provide information about the advantages and possible disadvantages of this technique.

## 5. Conclusions

Our research results were significant in terms of PT values in the Ex phase in patients who had undergone ACLR with the MAI technique. However, the fact that the Lyhsolm, Tegner, and IKDC scores revealed significant improvements in terms of improvement between pre–post tests and H/Q ratios revealed similar findings in ACLR-applied and healthy knees, revealing that the MAI ACLR technique can be used in terms of returning to sports within 6 months of the procedure. These results were supported by the similarity between the ACLR-applied knee and the HK in the JAPT, TPT, RD, and ER results in addition to the traditionally applied PT and H/Q ratios. In general, our research shows that when the advantages of the MAI technique are evaluated in terms of a return to sports, knee scores, and strength values, it is a viable method in sports surgery. Although the results of the present study reveal similarities between the ACLR-applied knee with the MAI technique and the healthy knee, the number of studies evaluating this technique is limited. In order to better evaluate the advantages and disadvantages of the MAI technique, studies involving short-, medium-, and long-term findings should be designed. In addition, the application of this technique in different subject groups such as athletes and sedentary people will provide more precise information. Designing studies comparing the MAI technique with other ACLR techniques will help to reveal the strengths and weaknesses of this technique more clearly.

## Figures and Tables

**Figure 1 diagnostics-13-02787-f001:**
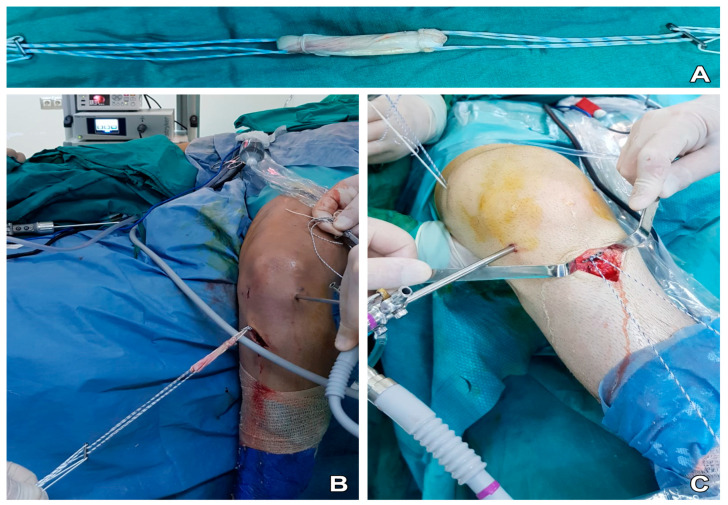
(**A**) Anatomically prepared single-bundle quadruple semitendinosus tendon autograft with adjustable suspension fixation, (**B**) placing the prepared graft into the knee, and (**C**) fixation of the graft in the tibial tunnel.

**Figure 2 diagnostics-13-02787-f002:**
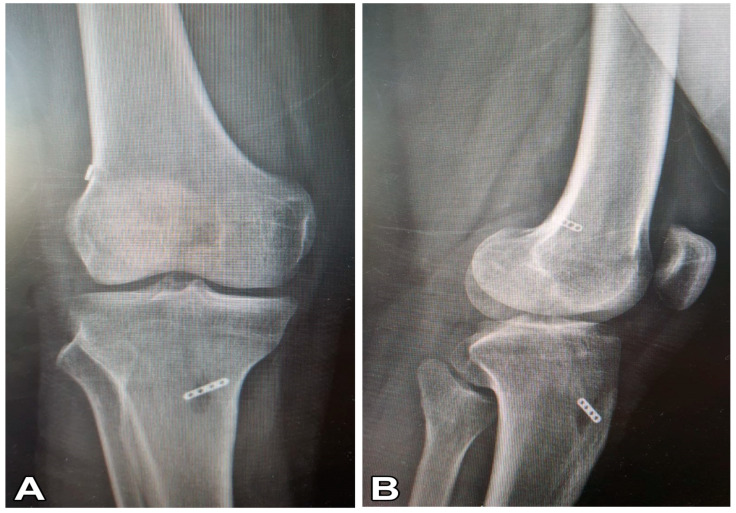
Post-operative roentgenography showing the use of the materials. (**A**) Post-op Knee anterior–posterior radiograph and (**B**) post-op knee lateral radiograph.

**Figure 3 diagnostics-13-02787-f003:**
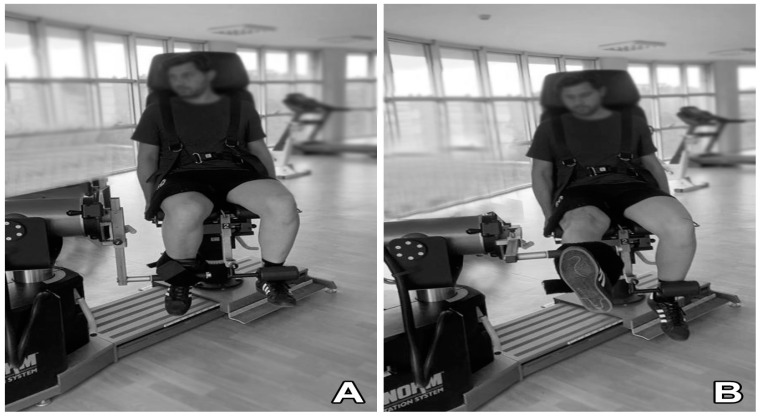
The knee flexion (**A**) and extension (**B**) movements by the isokinetic dynamometer.

**Table 1 diagnostics-13-02787-t001:** Characteristic and clinical variables of the study cohort (*n* = 20).

**Variables**	**Value**
Age, y	28.05 ± 6.87
Height, cm	176.75 ± 6.93
Weight, kg	86.40 ± 12.58
BMI, kg/m^2^	27.58 ± 2.83
Time from ACLR to measurements, mon	6.95 ± 0.83
**Dominant side**	**Value, n (%)**
Right	15 (75)
Left	5 (25)
**Surgical side**	**Value, n (%)**
Right	14 (70)
Left	6 (30)

Data are presented as mean ± SD or *n* (%). ACLR, anterior cruciate ligament reconstruction; BMI, body mass index.

**Table 2 diagnostics-13-02787-t002:** Pre- and post-operative functional knee outcome scores (Lysholm, IKDC, and Tegner).

	Pre-Operative		Post-Operative			
	Mean ± SD	Med (Min–Max)	Mean ± SD	Med (Min–Max)	95% CI	*p*
Lysholm	75.20 ± 8.13	75 (58–92)	98.20 ± 2.66	98.70 (90–100)	−26.99 to −20.01	<0.000 ^1^
IKDC	50.15 ± 8.80	50.15 (32–64)	91.25 ± 6.23	91.25 (80–100)	−45.79 to −36.41	<0.000 ^1^
Tegner	6.45 ± 1.19	6.45 (5–9)	6± 1.34	6 (4–8)	0.114 to 0.211	0.001 ^2^

*p* < 0.001; ^1^ Wilcoxon test; ^2^ Paired Samples *t*-test; 95% CI, pre–post confidence interval comparison results; IKDC, International Knee Documentation Committee; SD, standard deviation; Min, minimum; Max, maximum.

**Table 3 diagnostics-13-02787-t003:** Knee peak torque variables and the H/Q ratio post-operative reconstruction of the ACLR and HK (Mean ± SD).

Knee Peak Torque (Nm)	ACLR	HK	*t*	*p*-Value	ES	95% CI	
						LB	UB
60°/s Ex	183.55 ± 45.98	212.85 ± 40.90	−2.976	0.008	0.67	−49.91	−8.69
180°/s Ex	120.45 ± 29.99	134.80 ± 16.46	−2.460	0.024	0.59	−26.56	−2.14
240°/s Ex	108.50 ± 24.11	114.90 ± 17.18	−1.573	0.132	0.31	−14.92	2.12
300°/s Ex	82.15 ± 13.95	91.90 ± 17.82	−3.146	0.005	0.61	−16.24	−3.26
60°/s Flx	113.25 ± 29.42	120.10 ± 38.62	−1.293	0.212	0.20	−17.94	4.24
180°/s Flx	85.25 ± 23.64	86.70 ± 22.84	−0.348	0.732	0.06	−10.18	7.28
240°/s Flx	75.45 ± 15.38	76.65 ± 18.89	−0.358	0.725	0.07	−8.23	5.83
300°/s Flx	65.70 ± 11.75	64.05 ± 15.13	0.694	0.496	0.12	−3.33	6.63
**H/Q Ratio (%)**							
60°/s	63.60 ± 15.59	56.35 ± 14.21	1.762	0.094	0.49	−1.36	15.86
180°/s	71.75 ± 16.29	64.05 ± 13.22	1.696	0.106	0.52	−1.80	17.20
240°/s	70.40 ± 9.78	66.60 ± 12.80	1.301	0.209	0.33	−2.31	9.91
300°/s	80.50 ± 12.19	70.05 ± 12.76	3.535	0.002	0.84	4.26	16.64

*p* < 0.005; All comparisons had *p*-values < 0.05.; *t*, results of paired sample *t*-test; ES, effect size; 95% CI, pre–post confidence interval comparison results; LB, lower bound; UB, upper bound; HK, healthy; ACLR, Anterior cruciate ligament reconstruction; Ex, extension; Flx, flexion; Q, quadriceps; H, hamstring.

**Table 4 diagnostics-13-02787-t004:** The H/H and the Q/Q ratios post-operative reconstruction of the ACLR and HK (Mean ± SD).

Ratio (%)	H/H	Q/Q	*t*	*p*-Value	ES	95% CI	
						LB	UB
60°/s	106.46 ± 21.35	121.73 ± 35.03	−1.846	0.081	0.53	−32.58	2.04
180°/s	104.54 ± 22.20	117.35 ± 27.91	−1.664	0.113	0.51	−28.91	3.30
240°/s	102.51 ± 19.30	108.45 ± 17.18	−1.293	0.212	0.33	−15.56	3.68
300°/s	97.69 ± 16.26	112.77 ± 17.94	−3.583	0.002	0.88	−23.90	−6.27
*f*	0.717	1.002					
*p*	0.545	0.397					

*p* < 0.005; All comparisons had *p*-values < 0.05.; *t*, results of paired sample *t*-test; ES, effect size; 95% CI, pre–post confidence interval comparison results; LB, lower bound; UB, upper bound; HK, healthy; ACLR, Anterior cruciate ligament reconstruction; H/H, hamstring/hamstring ratio; Q/Q, quadriceps/quadriceps ratio.

**Table 5 diagnostics-13-02787-t005:** The comparison of JAPT values obtained in isokinetic tests for ACLR and HK sides (Mean ± SD).

JAPT (°)	ACLR	HK	*t*	*p*-Value	ES	95% CI	
						LB	UB
60°/s Ex	59.80 ± 5.47	60.55 ± 7.38	−0.424	0.676	0.12	−4.45	2.95
180°/s Ex	60.10 ± 8.23	58.75 ± 4.27	0.669	0.512	0.21	−2.88	5.58
240°/s Ex	61.70 ± 4.29	60.55 ± 4.19	0.856	0.403	0.27	−1.66	3.96
300°/s Ex	66.55 ± 5.62	59.55 ± 13.78	2.740	0.013	0.67	1.01	7.59
60°/s Flx	31.20 ± 6.72	33.70 ± 10.24	−1.145	0.267	0.29	−7.07	2.07
180°/s Flx	35.05 ± 6.96	39.25 ± 7.31	−2.115	0.048	0.59	−8.36	−0.04
240°/s Flx	37.30 ± 5.94	38.65 ± 9.86	−0.542	0.594	0.17	−6.56	3.86
300°/s Flx	31.40 ± 6.33	35.10 ± 7.41	−1.700	0.105	0.54	−8.26	0.86

All comparisons had *p*-values < 0.05.; *t*, results of paired sample *t*-test; ES, effect size; 95% CI, pre–post confidence interval comparison results; LB, lower bound; UB, upper bound; HK, healthy; ACLR, Anterior cruciate ligament reconstruction; Ex, extension; Flx, flexion; JAPT, joint angle at peak torque.

**Table 6 diagnostics-13-02787-t006:** The comparison of TPT values obtained in isokinetic tests for ACLR and HK sides (Mean ± SD).

TPT (s)	ACLR	HK	*t*	*p*-Value	ES	95% CI	
						LB	UB
60°/s Ex	0.76 ± 0.14	0.78 ± 0.11	−0.497	0.625	0.16	−0.09	0.06
180°/s Ex	0.32 ± 0.05	0.32 ± 0.03	−0.462	0.650	0	−0.02	0.02
240°/s Ex	0.25 ± 0.03	0.26 ± 0.03	−1.099	0.285	0.33	−0.02	0.01
300°/s Ex	0.20 ± 0.03	0.22 ± 0.03	−2.688	0.015	0.67	−0.04	−0.01
60°/s Flx	0.54 ± 0.12	0.61 ± 0.17	−2.039	0.056	0.48	−0.13	0.00
180°/s Flx	0.26 ± 0.04	0.29 ± 0.05	−2.067	0.053	0.66	−0.06	0.00
240°/s Flx	0.23 ± 0.04	0.23 ± 0.04	−0.264	0.795	0	−0.03	0.02
300°/s Flx	0.19 ± 0.04	0.20 ± 0.04	−1.541	0.140	0.25	−0.04	0.01

All comparisons had *p*-values < 0.05.; *t*, results of paired sample *t*-test; ES, effect size; 95% CI, pre–post confidence interval comparison results; LB, lower bound; UB, upper bound; HK, healthy; ACLR, Anterior cruciate ligament reconstruction; Ex, extension; Flx, flexion; TPT, time peak torque.

**Table 7 diagnostics-13-02787-t007:** The comparison of RD values obtained in isokinetic tests for ACLR and HK sides (Mean ± SD).

RD (s)	ACLR	HK	*t*	*p*-Value	ES	95% CI	
						LB	UB
60°/s Ex	0.13 ± 0.06	0.10 ± 0.04	1.532	0.142	0.59	−0.01	0.06
180°/s Ex	0.07 ± 0.03	0.06 ± 0.02	0.938	0.360	0.39	−0.01	0.02
240°/s Ex	0.05 ± 0.02	0.05 ± 0.01	0.256	0.800	0	−0.01	0.01
300°/s Ex	0.05 ± 0.02	0.05 ± 0.02	−0.396	0.697	0	−0.01	0.01
60°/s Flx	0.14 ± 0.06	0.14 ± 0.05	0.335	0.741	0	−0.03	0.05
180°/s Flx	0.08 ± 0.03	0.08 ± 0.04	−0.321	0.751	0	−0.03	0.02
240°/s Flx	0.06 ± 0.02	0.06 ± 0.02	−0.137	0.893	0	−0.01	0.01
300°/s Flx	0.05 ± 0.01	0.05 ± 0.01	−0.925	0.367	0	−0.01	0.00

All comparisons had *p*-values < 0.05.; *t*, results of paired sample *t*-test; ES, effect size; 95% CI, pre–post confidence interval comparison results; LB, lower bound; UB, upper bound; HK, healthy; ACLR, Anterior cruciate ligament reconstruction; Ex, extension; Flx, flexion; RD, reciprocal delay.

**Table 8 diagnostics-13-02787-t008:** The comparison of ER values obtained in isokinetic tests for ACLR and HK sides (Mean ± SD).

ER (Nm)	ACLR	HK	*t*	*p*-Value	ES	95% CI	
						LB	UB
240°/s Ex	85.85 ± 13.34	84.90 ± 16.55	0.207	0.838	0.06	−8.66	10.56
300°/s Ex	81.65 ± 9.27	78.00 ± 8.01	1.719	0.102	0.42	−0.79	8.09
240°/s Flx	90.95 ± 14.37	92.80 ± 25.56	−0.443	0.663	0.09	−10.59	6.89
300°/s Flx	87.55 ± 13.48	88.25 ± 14.79	−0.161	0.873	0.05	−9.78	8.38

All comparisons had *p*-values < 0.05.; *t*, results of paired sample *t*-test; ES, effect size; 95% CI, pre–post confidence interval comparison results; LB, lower bound; UB, upper bound; HK, healthy; ACLR, Anterior cruciate ligament reconstruction; Ex, extension; Flx, flexion; ER, endurance ratio.

## Data Availability

The datasets used and/or analyzed during the current study are available from the corresponding author upon reasonable request.

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
