# Peer review of "Post-Operative Modified All-Inside ACL Reconstruction Technique’s Clinical Outcomes and Isokinetic Strength Assessments"

_diagnostics, 2023, doi:10.3390/diagnostics13172787_

Round 1

Reviewer 1 Report

diagnostics-2464565_review

Title: Post-Operative Modified All-Inside ACL Reconstruction Technique's Clinical Outcomes and Isokinetic Strength Assessments

Comments for Authors

Dear authors,

I have carefully read your paper, which evaluated the Anterior cruciate ligament reconstruction (ACLR) with the modified all-inside 4ST grafting technique at the sixth post-operative month and analysed data related to isokinetic strength values, strength-related asymmetry rates, time parameters and joint angle in twenty athletes comparing the healthy knee and the side where the surgery was applied.

In your results you observed an improvement in the Lysholm, Tegner, and International Knee Documentation Committee scores after surgery in the operated side compared with preoperative levels. However, your results did not reveal a significant difference between the healthy knee and ACLR knee regarding strength, particularly in the flexion phase, and that similar findings were found in hamstring/quadriceps ratio parameters. Therefore, it appears that this ACLR technique can be used in athletes in view of strength gain and return to sports. In my opinion, more studies are needed to analyse in depth the effects of this technique with comparison of different ACLR techniques, a larger sample size and a longer follow-up time to support these results.

In general, the manuscript is well-written. The text is understandable and organized. However, I found some issues in abstract, introduction, methods, results, discussion and conclusion sections that should be addressed to improve the paper, in my opinion.

Specific comments:

Abstract

-     Page 1, line 27. I suggest you I suggest you mention that you evaluated the Lysholm, Tegner, and International Knee Documentation Committee in this section.

-     Page 1, line 35. H/Q is referring to hamstring/quadriceps, but what do the acronym HH/QQ mean? I suggest you add the full term I suggest you remove the phrase " in comparison with people received the usual care" as the results do not show these data on the between-group analysis.

-     Page 1, lines 38-42. Please rephrase the conclusions of the abstract to match what you have described in the conclusions section of the manuscript

Introduction

-     Page 2, lines 46-50. I suggest you add more information about the ACL pathology, age ranges in which it is more frequent, prevalence according to sex, etc.

-     Page 2, lines 59-60. Please, could you add more information about all-inside technique? I.e. list the advantages and disadvantages

-     Page 2, line 68. Flx/Ex, it is the first time that these terms are mentioned in the text, please add the full term before its abbreviation.

-     Page 2, line 82. PT, it is the first time that these terms are mentioned in the text, please add the full term before its abbreviation.

-     Page 2, lines 82-83. You have specified the hypothesis of your study. However, I suggest that you reformulate it in a more precise way.

Methods

- Page 3, lines 94-95. Please, could you add information about the places in which the data were recorded? City or Country? Where were the participants evaluated? Did the participants complete the questionnaires at home, at the university, hospital...? How long did it take to complete the entire assessment?

- Page 3, lines 100-101. You mentioned that GPower program was used to determine the number of participants. But how was sample size determined? Are based on previous studies? if so, add the necessary references. What is the statistical power? Please, add this information.

- Page 3, line 102. Why did you only include male in your sample? please add information, for example in the introduction or discussion that justifies your decision. In the same way they should add this as a limitation of their study

- Page 3, lines 105-106. Please, add a brief description of the Lysholm knee scale, International Knee Documentation Committee and Tegner Activity Scale, also add the necessary references to support this information.

- Page 5, line 152. Grammar mistake, “300o”, please check it.

- Page 5, lines 154 and 156.  You mentioned “warm-up protocol” and fixed protocol”. Please add references to support this information.

Results

-     Page 6, line 199. The information that appears in the results section is disordered; it makes difficult read the manuscript. I suggest you rearrange it. You could add a title to introduce subsections. First explain the results you want to comment on or highlight from the tables and then show the corresponding table.

-     Table 2: Add in the table footer when the p value is significant, as you mentioned in your methods sections: p<0.05

-     Tables 3-8. You mention in your statistical analysis section thatThe statistical results were evaluated within 197 a significance level of p<0.05”. However, you put in the footer of the tables 3-8,All comparisons had P-values ≥0.05” please check this information.

-     Table 4. The interpretation of table 4 is confusing, information is missing, you do not specify which data refer to the operated knee and which to the healthy knee. There is only one column for H/H data and one column for Q/Q data. I suggest that you please express the data as they appear organized in Table 3 or separate the information for H/H ratio post-operative reconstruction of the ACLR and HK in one table and Q/Q ratio post-operative reconstruction of the ACLR and HK in another table.

-     You have added information about the effect size. Your results show a moderate and even high effect size for some of the variables, although in the comparative analysis it does not show significant results. I consider that this information is relevant in the interpretation of your results, I suggest that you include it to show and comment on your results in a more complete and precise way.

-     Discussion

-     Page 11. Please add the limitations of your study in this section.

Conclusions

-     Page 11.  I suggest you to the reformulate your conclusions in a more carefully way such as: “Our results suggest that ….”

I hope that my comments could help to improve the paper.

 Minor editing of English language required

Author Response

Dear authors,

I have carefully read your paper, which evaluated the Anterior cruciate ligament reconstruction (ACLR) with the modified all-inside 4ST grafting technique at the sixth post-operative month and analysed data related to isokinetic strength values, strength-related asymmetry rates, time parameters and joint angle in twenty athletes comparing the healthy knee and the side where the surgery was applied.

In your results you observed an improvement in the Lysholm, Tegner, and International Knee Documentation Committee scores after surgery in the operated side compared with preoperative levels. However, your results did not reveal a significant difference between the healthy knee and ACLR knee regarding strength, particularly in the flexion phase, and that similar findings were found in hamstring/quadriceps ratio parameters. Therefore, it appears that this ACLR technique can be used in athletes in view of strength gain and return to sports. In my opinion, more studies are needed to analyse in depth the effects of this technique with comparison of different ACLR techniques, a larger sample size and a longer follow-up time to support these results.

In general, the manuscript is well-written. The text is understandable and organized. However, I found some issues in abstract, introduction, methods, results, discussion and conclusion sections that should be addressed to improve the paper, in my opinion.

Specific comments:

Abstract

-     Page 1, line 27. I suggest you I suggest you mention that you evaluated the Lysholm, Tegner, and International Knee Documentation Committee in this section.

Response: Thanks for your review. Added to the abstract section where Lysholm, Tegner, and International Knee Documentation Committee scores are obtained.

-     Page 1, line 35. H/Q is referring to hamstring/quadriceps, but what do the acronym HH/QQ mean? I suggest you add the full term I suggest you remove the phrase " in comparison with people received the usual care" as the results do not show these data on the between-group analysis.

Response: As you said, we added the HH and QQ expansions where you want them. However, we could not find the sentence you mentioned as "in comparison with people received the usual care" in the article.

-     Page 1, lines 38-42. Please rephrase the conclusions of the abstract to match what you have described in the conclusions section of the manuscript

Response: Thanks for your review. Conclusions of the abstract were rephrased to match the conclusions section of the article.

Introduction

-     Page 2, lines 46-50. I suggest you add more information about the ACL pathology, age ranges in which it is more frequent, prevalence according to sex, etc.

Response: Thanks for your review. Added more detailed information about ACL.

-     Page 2, lines 59-60. Please, could you add more information about all-inside technique? I.e. list the advantages and disadvantages

Response: Thanks for your review. Added more information about all-inside technique such as advantages and disadvantages.

-     Page 2, line 68. Flx/Ex, it is the first time that these terms are mentioned in the text, please add the full term before its abbreviation.

Response: Thanks for your review. Added the full term before its abbreviation.

-     Page 2, line 82. PT, it is the first time that these terms are mentioned in the text, please add the full term before its abbreviation.

Response: Thanks for your review. Added the full term before its abbreviation.

-     Page 2, lines 82-83. You have specified the hypothesis of your study. However, I suggest that you reformulate it in a more precise way.

Response: Paragraph separated and main hypothesis added.

Methods

- Page 3, lines 94-95. Please, could you add information about the places in which the data were recorded? City or Country? Where were the participants evaluated? Did the participants complete the questionnaires at home, at the university, hospital...? How long did it take to complete the entire assessment?

Response: Necessary information added.

- Page 3, lines 100-101. You mentioned that GPower program was used to determine the number of participants. But how was sample size determined? Are based on previous studies? if so, add the necessary references. What is the statistical power? Please, add this information.

Response: Thanks for your contribution. As you said, we have added the gpower result and the sample study we used for the result to the relevant section.

- Page 3, line 102. Why did you only include male in your sample? please add information, for example in the introduction or discussion that justifies your decision. In the same way they should add this as a limitation of their study

Response: For the results to be valid, only the male population could qualify. The number of our female patients is valuable enough. This information has been added to the limitation.

- Page 3, lines 105-106. Please, add a brief description of the Lysholm knee scale, International Knee Documentation Committee and Tegner Activity Scale, also add the necessary references to support this information.

Response: Descriptive information for relevant scales added with references.

- Page 5, line 152. Grammar mistake, “300o”, please check it.

Response: Grammar correction made.

- Page 5, lines 154 and 156.  You mentioned “warm-up protocol” and fixed protocol”. Please add references to support this information.

Response: References added.

Results

-     Page 6, line 199. The information that appears in the results section is disordered; it makes difficult read the manuscript. I suggest you rearrange it. You could add a title to introduce subsections. First explain the results you want to comment on or highlight from the tables and then show the corresponding table.

Response: The results section has been rearranged.

-     Table 2: Add in the table footer when the p value is significant, as you mentioned in your methods sections: p<0.05

Response: Added the terms "p<0.05 or p<0.01" to the table footer when the p value is significant.

-     Tables 3-8. You mention in your statistical analysis section that “The statistical results were evaluated within 197 a significance level of p<0.05”. However, you put in the footer of the tables 3-8, “All comparisons had P-values ≥0.05” please check this information.

Response: This information has been checked and corrected.

-     Table 4. The interpretation of table 4 is confusing, information is missing, you do not specify which data refer to the operated knee and which to the healthy knee. There is only one column for H/H data and one column for Q/Q data. I suggest that you please express the data as they appear organized in Table 3 or separate the information for H/H ratio post-operative reconstruction of the ACLR and HK in one table and Q/Q ratio post-operative reconstruction of the ACLR and HK in another table.

Response: First of all, thanks for your detailed review. However, as you know, the Q/Q and H/H ratios are not comparable between ACLR and healthy sides. Since the Q/Q ratio is the ratio of the Q muscle strength on the ACLR side and the Q muscle strength on the healthy side. This is also true for H. Here, the comparison of Q/Q and H/H ratios at different angular velocities with the t-test includes post-operative bilateral evaluation of the healthy and ACLR sides of the patients. In addition, f and p values in the lower rows include comparisons of H/H and Q/Q ratios at all angular velocities. Despite our explanations, if you still have any revision requests, we are ready to do so. Thanks again.

-     You have added information about the effect size. Your results show a moderate and even high effect size for some of the variables, although in the comparative analysis it does not show significant results. I consider that this information is relevant in the interpretation of your results, I suggest that you include it to show and comment on your results in a more complete and precise way.

Response: Added comment about effect size.

-     Discussion

-     Page 11. Please add the limitations of your study in this section.

Response: Added limitation to discussion section.

Conclusions

-     Page 11.  I suggest you to the reformulate your conclusions in a more carefully way such as: “Our results suggest that ….”

Response: In the conclusions section, the results are explained in more detail.

I hope that my comments could help to improve the paper.

Reviewer 2 Report

This article presents an intriguing study, and I have several suggestions for improvement:

1.          In the abstract, refrain from using unexplained abbreviations such as 4ST, HK, IKDC, H/Q, and HH/QQ.

2.          Consider using a more specific keyword instead of "athletes."

3.          In line 47, when describing ACL, consider incorporating the following up-to-date references:

https://www.mdpi.com/2075-4426/13/7/1022

https://www.mdpi.com/2227-9059/11/2/507

4.          For readers unfamiliar with anthropometric data interpretation, please provide more detailed definitions of RD, JAPT, TPT, and ER.

5.          Define PT in line 82, as it appears for the first time in the manuscript.

6.          In lines 100-101, provide additional parameters regarding GPower sample size determination. What inputs led to a sample size of 20?

7.          Clarify the meaning of "malleus" in line 160. Malleus is a small bone in the ear. Do you mean "malleolus" instead?

8.          In line 193, if there were no >=3 group comparisons in the study, the ANOVA test might not be necessary.

9.          Specify the definition of "dominant knee" in Table 1.

10.      Please enhance the clarity of the explanation for the 95% CI in Table 2. Specifically, could you clarify whether the 95% CI pertains to the difference in the three selected functional scores before and after the operation?

11.      Explain what "t" represents in Table 3.

12.      In Table 4, elaborate on the clinical implications of H/H and Q/Q, as well as their definitions.

13.      For Tables 6, 7, and 8, provide explanations for all columns. I assume the unexplained columns are for "t."

I hope that implementing these suggestions will enhance the clarity and comprehensibility of your manuscript. Thank you for your valuable contribution to the field of research.

The manuscript's English usage appears to be from non-native speakers. It is advisable to consider utilizing chatGPT to improve the proficiency and refinement of the English language in the manuscript.

Author Response

  1. In the abstract, refrain from using unexplained abbreviations such as 4ST, HK, IKDC, H/Q, and HH/QQ.

Response: Thanks for your suggestion. Another reviewer said that the full terms of the abbreviations should be written. Since we did not exceed the word limit, we used abbreviations with their full terms.

  1. Consider using a more specific keyword instead of "athletes."

Response: We removed "athletes" from the keywords. We did not add any extra keywords in its place.

  1. In line 47, when describing ACL, consider incorporating the following up-to-date references:

https://www.mdpi.com/2075-4426/13/7/1022

https://www.mdpi.com/2227-9059/11/2/507

Response: Thanks for your contribution. Added as references 4 and 7.

  1. For readers unfamiliar with anthropometric data interpretation, please provide more detailed definitions of RD, JAPT, TPT, and ER.
  2. Define PT in line 82, as it appears for the first time in the manuscript.

Response: Thanks for your review. Added the full term before its abbreviation.

  1. In lines 100-101, provide additional parameters regarding GPower sample size determination. What inputs led to a sample size of 20?

Response: Thanks for your contribution. As you said, we have added the gpower result and the sample study we used for the result to the relevant section.

  1. Clarify the meaning of "malleus" in line 160. Malleus is a small bone in the ear. Do you mean "malleolus" instead?

Response: Thanks for your suggestion. Necessary correction has been made.

  1. In line 193, if there were no >=3 group comparisons in the study, the ANOVA test might not be necessary.

Response: The ANOVA test was used for the internal analysis of H/H and Q/Q ratios at all angular velocities. Updated in statistical analysis section.

  1. Specify the definition of "dominant knee" in Table 1.

Response: We revised Table 1 and provided the necessary explanations as comments under the table. If there is an incomprehensible situation, we are ready to revise it again.

  1. Please enhance the clarity of the explanation for the 95% CI in Table 2. Specifically, could you clarify whether the 95% CI pertains to the difference in the three selected functional scores before and after the operation?

Response: Thank you for your suggestion. But we couldn't understand exactly what you want. 95% CI was within negative ranges for Lyhsholm and IKDC and positive for Tegner. When statistical analysis scores are evaluated in terms of 95% CI, "-" or "+" direction without passing the "0" point are important results for observing the significance. Here, "-" for two parameters and "+" for one parameter changed depending on the score values of the parameters. If you give us detailed information about this issue, we are ready to fix it. However, we could only give this answer in this situation. Thank you very much for your contribution.

  1. Explain what "t" represents in Table 3.

Response: Added what the t value is to the table descriptions.

  1. In Table 4, elaborate on the clinical implications of H/H and Q/Q, as well as their definitions.

Response: Added full wording of H/H and Q/Q abbreviations to table description.

  1. For Tables 6, 7, and 8, provide explanations for all columns. I assume the unexplained columns are for "t."

Response: Added what the t value is to the table descriptions.

I hope that implementing these suggestions will enhance the clarity and comprehensibility of your manuscript. Thank you for your valuable contribution to the field of research.

Comments on the Quality of English Language

The manuscript's English usage appears to be from non-native speakers. It is advisable to consider utilizing chatGPT to improve the proficiency and refinement of the English language in the manuscript.

Thank you for your valuable contrubition.

Best Regards.

Reviewer 3 Report

Dear authors,

In this paper, you sought to assess knees which underwent ACLR with MAI technique and 85 healthy knees using additional parameters obtained with isokinetic tests in addition to 86 traditional data such as PT and H/Q. Overall, I believe that the presentation quality and language use does not meet the requirements of the Journal and further work is required by the authors to reach published standard.

Please find specific comments below

Title: The study design needs to be mentioned in the title of the paper.

Abstract

Please note that reporting p values is not sufficient. 

Although utilizing GPower 3.1 software is an acceptable way to calculate the appropriate number of participants/sample size, I would advise you to provide the readers with more details to achieve more transparency in your reporting.

Table 1

Heigh and weigh should be height and weight.

Discussion section

Please note this section needs to be more appropriately organised. In addition you will need to increase  it in length by adding a further study limitations / implications for future research section.

Major revision is needed.

Author Response

Dear authors,

In this paper, you sought to assess knees which underwent ACLR with MAI technique and 85 healthy knees using additional parameters obtained with isokinetic tests in addition to 86 traditional data such as PT and H/Q. Overall, I believe that the presentation quality and language use does not meet the requirements of the Journal and further work is required by the authors to reach published standard.

Please find specific comments below

Title: The study design needs to be mentioned in the title of the paper.

Abstract

Please note that reporting p values is not sufficient. 

Although utilizing GPower 3.1 software is an acceptable way to calculate the appropriate number of participants/sample size, I would advise you to provide the readers with more details to achieve more transparency in your reporting.

Response: Thanks for your contribution. As you said, we have added the gpower result and the sample study we used for the result to the relevant section.

Table 1

Heigh and weigh should be height and weight.

Response: Correction made.

Discussion section

Please note this section needs to be more appropriately organised. In addition you will need to increase  it in length by adding a further study limitations / implications for future research section.

Response: Thanks for your contributions. We tried to reorganize the discussion section. We also added the main and major limitations of our research.

Thank you for ypur valuable contribution

Best Regards.

Round 2

Reviewer 2 Report

Thanks for the revision. 

1. Please clarify in each table what 95% CI means. Take table 2 for example, 95% CI means 95% CI for Pre-operative ? Post-Operative ? Pre - post ( I guess) ? Post - pre ?  Why not use post- pre but use pre-post instead ? In addition, what did " %95 CI" mean in table 3-8 ?

2. Please offer an simple illustration to explain the anthropometric parameters for the readers not familiar with these parameters. These parameters could be added to the keywords in the abstract section. 

3. Please define the t values in the section of statistical analyses. 

4. Describe for about the clinical implication of your study. After this study, you will suggest more or suggest less about MAI to your colleagues ?

Moderate English editing is necessary. 

Author Response

Thanks for the revision. 

Response: Thank you for your contribution to our research. We tried to make all the corrections you mentioned. We are ready to make any extra revision requests you have.

Please clarify in each table what 95% CI means. Take table 2 for example, 95% CI means 95% CI for Pre-operative ? Post-Operative ? Pre - post ( I guess) ? Post - pre ?  Why not use post- pre but use pre-post instead ? In addition, what did " %95 CI" mean in table 3-8 ?

Response: As you said, we have added that 95% CI values reflect the pre-post results at the bottom of all tables. We also added the statistical analysis section. Thanks for your contribution.

Please offer an simple illustration to explain the anthropometric parameters for the readers not familiar with these parameters. These parameters could be added to the keywords in the abstract section. 

Response: As we understand it, we have added a few new keywords to reach readers. Thanks for your contribution

Please define the t values in the section of statistical analyses. 

Response: In the statistical analysis part, it was added that the "t" value was applied for HK-ACLR and pre-post tests. Thanks for your contribution.

Describe for about the clinical implication of your study. After this study, you will suggest more or suggest less about MAI to your colleagues ?

Response: As you said, we mentioned in the conclusion part that the MAI technique is a method that can be used by sports surgeons in terms of the results. Thanks for your contribution.

Reviewer 3 Report

Dear authors,

I believe you have now succesfully addressed my original comments.

with kind regards 

Author Response

Dear authors,

I believe you have now succesfully addressed my original comments.

with kind regards 

Response: Dear reviewer, We are grateful for your contribution to our research.

With kind regards.
